# Mapping the prevalence and covariates associated with home delivery in Bangladesh: A multilevel regression analysis

**Rakhi Dey**[1], **Susmita Rani Dey**[1], **Meem Haque**[1], **Anushuya Binta Rahman**[1], **Satyajit Kundu**[2], **Sarmistha Paul Setu**[1], **U. K. Majumder**[1]*

1 Statistics Discipline, Khulna University, Khulna, Bangladesh, 2 School of Medicine and Dentistry, Griffith University, Gold Coast, Australia

* majumderuk@ku.ac.bd

## Abstract

### Introduction

Bangladesh has made an intense effort to improve maternal healthcare facilities including facility delivery, but the number of home deliveries is still very high. Therefore, this study aims to find out district-wise prevalence and determine the individual and community-level predictors of home delivery among women in Bangladesh.

### Methods

Data were derived from the Multiple Indicator Cluster Survey (MICS) 2019, a nationwide cross-sectional survey in Bangladesh. A final sample of 9,166 (weighted) women who gave birth in the two years preceding the survey were included in this study. Considering the two-stage cluster sampling strategy adopted by MICS, we used multilevel (2-level) logistic regression analysis to find out the correlates of home delivery.

### Results

The overall weighted prevalence of home delivery was 46.41% (95% confidence interval [CI]: 45.39–47.43). The highest prevalence was observed in Bandarban district (84.58%), while the lowest was found in Meherpur district (6.95%). The intercept-only regression model demonstrates that the likelihood of women from various clusters having home delivery varied significantly (variance: 1.47, standard error [SE]: 0.117), indicating the applicability of multilevel regression modeling. The multilevel regression analysis showed that women with higher education, wealth status and ANC visit, and those aged >18 years at first marriage/union were associated with lower odds of delivering child at home compared to their counterparts. While women from age group of 35–49 years, whose last pregnancy was unintended were more likely to deliver child at home. In addition, those respondents belonging to a community that had higher wealth status, women's education level, and exposure to media showed lower odds of having delivery at home.

**Data Availability Statement:** All relevant data are within the paper and its Supporting information files.

**Funding:** The author(s) received no specific funding for this work.

**Competing interests:** The authors have declared no competing interests exists.

## Conclusions

The finding indicates that delivery at home is still high in Bangladesh. Targeted interventions to reduce home delivery are urgently needed in Bangladesh to tackle adversities during deliveries and save mothers from the consequences.

## Background

In underdeveloped nations, home deliveries have been shown to have unfavorable effects, despite the ongoing discussion in wealthy nations regarding well-being and women's rights to select between home and institutional birth [1–3]. It is typical in wealthy nations to presume that women and newborns should get hospital treatment during birth [1, 4]. In most, but not all, nations during the past few decades, there has been a marked decline in home births [5–7]. The expansion of institutional delivery coverage and the use of skilled birth attendants during deliveries are only a couple of the measures that have been put forth to lower maternal, fetal, and newborn mortality [8, 9].

According to the World Bank, Bangladesh attained a significant decline in maternal death between 1990 and 2017. The maternal mortality ratio (MMR) in Bangladesh decreased from 574 deaths per 100,000 live births in 1990 to 173 deaths per 100,000 live births in 2017. This represents a considerable reduction, although challenges remain in further reducing the MMR [10]. But as stated by the World Health Organization (WHO), an estimated 295,000 women faced death due to pregnancy-related reasons in 2017 [11]. In 2020, about 800 women per day died from gestation and delivery-related avoidable reasons [12]. The maternal mortality rate is very high nowadays. In 2020, there were over 2,87,000 deaths of women during and after pregnancy and delivery. In low and lower-middle-income nations, around 95% of all maternal deaths occurred in 2020, the majority could have been avoided [13]. By ensuring that there is emergency delivery care available when needed and advanced surveillance, it is possible to avoid more than 40% of stillbirths that occur at the moment of delivery [14, 15].

The Millennium Development Goals (MDG) and the Sustainable Development Goals (SDG) were the first global goals and targets that attempted to establish, measure, and attain global progress in health and development before the turn of the era [16]. Reducing maternal death is a worldwide precedence, and it is one of the targets of the United Nations' SDG. Target 3.1 of SDG aims to reduce the ratio of maternal death to less than 70 per 100,000 live births within 2030 [17]. Efforts to achieve this target involve improving access to maternal healthcare, ensuring skilled attendants during childbirth, promoting family planning, strengthening health systems, and addressing the social and economic factors that contribute to maternal deaths [17]. Due to factors that are often avoidable, the majority of these fatalities (99%) and complications happen in low- and middle-income countries [14, 18]. The discussion about the ideal location for delivery is frequently more emotional than fact-based because there haven't been many studies that carefully compare home versus hospital deliveries. However, little is known about the long-term effects of planned or unexpected home deliveries [3].

Previous literature suggests that women's education and employment, pregnancy intention, religious belief, media exposure, wealth status, place of residence, ANC visits, and having living children were significantly associated with home delivery [19–22]. According to a qualitative study conducted in Bangladesh, the main reason why women prefer home delivery is poverty. Other factors that may contribute to this preference include religious fallacies, traditional beliefs, bad road conditions, and the inability of women to participate in family decision-

making, and a lack of transportation to the closest medical facility [23]. Another research conducted in Bangladesh among urban women found that 36.5% of urban women gave birth at home and that women from wealthier households and those who had more antenatal care (ANC) visits were less likely to do so [20].

Bangladesh has implemented several initiatives related to maternal health services to address maternal health. The key maternal health services available in Bangladesh include antennal care, post-abortion care, basic essential obstetric care (EOC), comprehensive EOC, postnatal care for mothers, intensive care unit for mothers, introduction of health vouchers scheme for poor women, deployment of community-based skilled birth attendants, and introduction of the midwifery program [24–26]. But not all health facilities, particularly those in rural regions in Bangladesh, offer all kinds of maternity and newborn health services [25]. Consequently, despite progress, Bangladesh still faces challenges in reducing maternal mortality [27]. Issues such as inadequate healthcare infrastructure, geographical barriers, limited access to skilled birth attendants, and socioeconomic disparities in Bangladesh continue to affect maternal health outcomes [27, 28]. However, most studies in Bangladesh focused on either facility delivery or a sub-sample of women having home delivery, and there hasn't been much research looking at the variables related to home delivery in Bangladesh using country representative data, despite the country's high prevalence of home deliveries and the corresponding high rates of maternal and infant death. Therefore, our study's findings help in identifying the factors influencing home delivery in Bangladesh using a nationally representative sample.

## Methods

### Study design and participants

Data from a cross-sectional survey, Multiple Indicator Cluster Survey (MICS) was obtained. The survey was directed at the household (HH) level, where data were collected from 64 districts in Bangladesh. Data from the households were gathered by applying a two-stage stratified cluster sampling technique to guarantee national representation. The enumeration areas (EAs) from the last census in Bangladesh were considered as the primary sampling unit (PSU). A sample of 20 households was taken from each PSU systematically. Finally, a total of 3,220 PSUs yielded a total sample of 64,400 households. The detailed information on the sampling technique, questionnaire, and study procedure can be found in the MICS 2019 report [29]. Women's data file was used in this investigation where a total of 64,870 eligible ever-married reproductive-aged women aged between 15 and 49 years were interviewed. The ever-married women were those who had been married at least once in their lives though they may not be currently married [30]. After excluding all missing cases, the final analysis included a total number of 9,166 (weighted) women who gave birth in the two years preceding the survey. The sample selection and case exclusion from MICS 2019 has been shown in Fig 1.

### Data source

Our study analyzed nationally representative data from the MICS 2019 in Bangladesh. The Bangladesh Bureau of Statistics (BBS) and UNICEF collaborated to undertake a six-round worldwide MICS. MICS has been considered as the key source of trustworthy statistical evidence on women and children globally through a face-to-face interview method directed by skilled field workers. MICS covers a wide range of themes including information on maternal and child health through household survey [29].

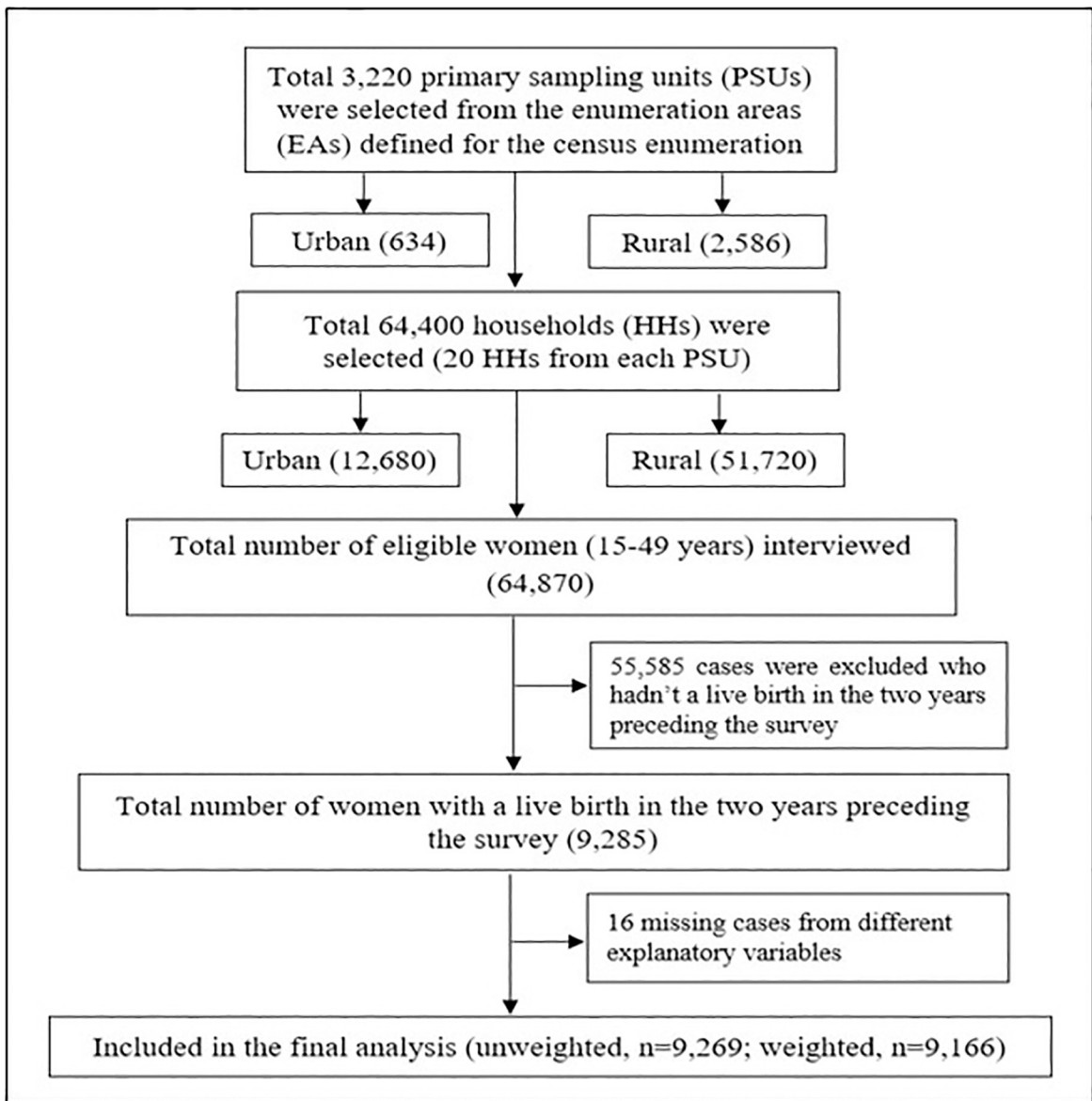

**Fig 1. Flow chart of the participants' selection from MICS 2019 data (women file).**

### Response variable

In our study, "place of delivery" was the response variable which was measured using the question "*Where did you give birth to [Name of the child]*?" The outcome variable was then dichotomized and recoded as '1' for home delivery, and '0' for facility-based delivery. 'Home delivery' was considered when the women gave birth at their own home or other's home, and when the birth was at any health facility setting, it was considered as 'facility-based delivery'. The other's home referred to the home of neighbors, friends or relatives of the respondents, or any birth attendant's home [31].

## Predictor variables

Different individual factors as well as community-level factors were scrutinized as predictor variables for employing the multilevel modeling. We included age of women (15–19, 20–34, 35–49 years), women's educational qualification (pre-primary/no formal education, primary, secondary, higher secondary+), age at first marriage/union (≤18 years, >18 years), wealth status (poorest, second, middle, fourth, richest), number of ANC visit (no visit, 1–3 visits, 4–7 visits, 8 visits or above), last pregnancy intension (intended, unintended), and exposure to media (no, yes) as individual-level variables. The household wealth status was calculated based on the ownership of different household assets using principal component analysis (PCA) [32]. Media exposure was created as a dichotomous variable. Women were categorized as having media exposure if they reported access to at least one of the following information sources like reading newspapers/magazines, listening radio, or watching television. Those with access to any of these sources were classified as 'yes' for media exposure, while those without access to any were classified as 'no' [29]. All of these predictor variables were selected after reviewing previous related literature [19, 21–23]. In addition, type of place of residence (rural, and urban), administrative division (Barishal, Chattogram, Dhaka, Khulna, Mymenshingh, Rajshahi, Rangpur, and Sylhet), community-level wealth status (whether or not the cluster's top three wealth quintiles included more than 50% of respondents), Community-level women education (whether more than 50% of respondents in the cluster had at least a secondary education or less education, up to primary level), and community-level media exposure (if more than 50% of respondents in the cluster have access to the media or not) [33, 34].

## Data analysis

In this study, we used descriptive statistics to present the basic features of respondents and the distribution of home delivery across different categories of the variables. Bivariate association between home delivery and other explanatory variables was tested using Pearson chi-square analysis. After allocating sample weight, utilizing clusters as the primary sampling unit (PSU), and stratifying the sample, weighted calculations were performed using the "svy" command for all descriptive and bivariate analyses. Additionally, a nationwide map is depicted to show the district-level distribution of home delivery in Bangladesh. Considering the complex sampling strategy (hierarchical) adopted by MICS, multilevel (2-level) logistic regression analysis was employed to find out the correlates of home delivery after adjusting the cluster effects [35]. For multilevel modeling, we constructed four regression models (Model 0 to Model 3). The intercept-only model (null model) was denoted in Model 0 without including any predictor to estimate the cluster-level variance in the outcome variable. Individual-level factors were the focus of Model 1, while community-level variables were incorporated into Model 2. Every explanatory variable both at individual and community level, was incorporated into the final model (Model 3). For all regression models, we regarded the clusters as level-2 factors. Prior to constructing the regression models, multi-collinearity among the explanatory variables was examined using the variance inflation factor (VIF). After employing the multilevel models, the intra-class correlation coefficient (ICC) was used to measure the community variation. Additionally, the median odds ratio (MOR) and proportionate change in variance (PCV) were utilized as indices of variation [36]. Akaike information criterion (AIC) was estimated to test the model fitness. The degree and intensity of association between the response and the predictors were determined using the adjusted odds ratio (AOR) and 95% confidence interval (CI). Statistical significance was considered at 5% level (p<0.05). Stata (version 16.0) was used for all of the statistical analyses, and ArcGIS (version 10.8) was used to create the map.

### Ethics approval

This study did not require any ethical approval as the analysis used only de-identified existing unit record data from the secondary data source MICS.

## Results

### Background characteristics of the participants

A total of 9,166 (weighted) women who had at least one live delivery in the two years preceding the survey were included in this study. Most of the women were from the age group of 20–34 years (77.13%) and about 9.2% of women had no formal education. Around 72% of women have had their first marriage/union at below or equal to 18 years. Regarding the ANC visit, 17.20% of women didn't receive any ANC, while 36.89% of women received 4 or above ANC, with only 4.88% having WHO-recommended 8 or above ANC visits. About 25% of the children were unwanted, and almost 35% of women didn't have any exposure to media. In this study, a large number of women were from rural areas (78.07%), and the highest number was from the Dhaka division (24.16%) and the least from the Barishal division (5.53%) (Table 1).

### Prevalence and bivariate distribution of home delivery

The overall prevalence of home delivery in Bangladesh was 46.41% (95% CI: 45.39%–47.43%). Table 1 displays the percentage of women who used home delivery by both individual and community-level variables. A significant difference in having home delivery across different categories of the explanatory variables was found, and all the explanatory variables showed a significant association in the bivariate distribution of delivery place (all p<0.05). While looking at the district-level prevalence, the peripheral districts had a higher proportion of women who had home delivery. Home delivery was least common in Meherpur (6.95%), followed by Rajshahi (19.33%) and Chuadanga district (20.30%) in Bangladesh, and most common in Bandarban (84.58%), Sherpur (82.02%), and Khagrachari district (76.06%) (Fig 2).

### Factors associated with home delivery

**Measures of variation (random-effects).** The intercept-only regression model (Model 0) indicated that the likelihood of women from various clusters using home birth varied significantly (variance: 1.47, SE: 0.117). The ICC value of Model 0 suggested that 30.8% of the total variation in using home delivery was a result of differences from cluster to cluster. Based on the model-fitness statistics, we selected Model 4 as our final model to interpret the findings. Significant variations were found in the final model (Model 3), and the impact of community heterogeneity was shown by the MOR of 1.84. It implies that a woman's chances of utilizing home delivery would rise by 1.84-fold on average if she relocated to a cluster where home deliveries are more common. Furthermore, the PCV shows that both community- and individual-level variables account for 72.11% of the variance in the probabilities of home delivery within communities (Table 2).

**Measures of associations (fixed-effects).** Compared to younger women, women in the 35–49 age range had a higher likelihood of giving birth at home (AOR: 1.29, 95% CI: 1.02–1.63). Compared to those with no formal education or pre-primary only, participants with at least secondary education and upper secondary+ education were 42% (AOR: 0.58, 95% CI: 0.46–0.72) and 69% (AOR: 0.31, 95% CI: 0.24–0.41) less likely to give birth at home, respectively. The likelihood of home delivery decreased as the household wealth index increased, and there was a substantial correlation between household wealth status and home delivery. The results also showed that respondents having at least 1–3 ANC visits (AOR: 0.38, CI: 0.33–0.45),

**Table 1. Bivariate distribution of home delivery by selected independent variables.**

| Variables | Total n (%) | Home delivery | | P value |
|---|---|---|---|---|
| | | No; n (%) | Yes; n (%) | |
| **Overall prevalence; % (95% CI)** | 9166 (100) | 53.59 (52.26–54.92) | 46.41 (45.08–47.74) | |
| **Individual-level characteristics** | | | | |
| **Age of women** | | | | <0.001 |
| 15–19 years | 1246 (13.59) | 675 (54.17) | 571 (45.83) | |
| 20–34 years | 7069 (77.13) | 3883 (54.92) | 3187 (45.08) | |
| 35–49 years | 851 (9.28) | 355 (41.69) | 496 (58.31) | |
| **Women's education level** | | | | <0.001 |
| Pre-primary or none | 841 (9.17) | 206 (24.47) | 635 (75.53) | |
| Primary | 2129 (23.23) | 762 (36.08) | 1361 (63.92) | |
| Secondary | 4587 (50.04) | 2638 (57.52) | 1949 (42.48) | |
| Higher secondary+ | 1609 (17.56) | 1300 (80.78) | 309 (19.22) | |
| **Wealth status** | | | | <0.001 |
| Poorest | 1948 (21.25) | 510 (26.18) | 1438 (73.82) | |
| Second | 1726 (18.83) | 718 (41.59) | 1008 (58.41) | |
| Middle | 1744 (19.02) | 943 (54.07) | 801 (45.93) | |
| Fourth | 1816 (19.81) | 1192 (65.63) | 624 (34.37) | |
| Richest | 1932 (21.08) | 1550 (80.2) | 383 (19.8) | |
| **Age at first marriage/union** | | | | <0.001 |
| ≤ 18 years | 6632 (72.35) | 3387 (51.08) | 3244 (48.92) | |
| >18 years | 2534 (27.65) | 1525 (60.16) | 1010 (39.84) | |
| **ANC visit** | | | | <0.001 |
| No visit | 1576 (17.20) | 307 (19.48) | 1269 (80.52) | |
| 1–3 visit | 4209 (45.92) | 2031 (48.25) | 2178 (51.75) | |
| 4–7 visit | 2934 (32.01) | 2186 (74.52) | 748 (25.48) | |
| 8 and above | 447 (4.88) | 388 (86.73) | 59.39 (13.27) | |
| **Last pregnancy intension** | | | | <0.001 |
| Intended | 6885 (75.12) | 3849 (55.89) | 3037 (44.11) | |
| Unintended | 2280 (24.88) | 1063 (46.63) | 1217 (53.37) | |
| **Exposure to media** | | | | <0.001 |
| No | 3181 (34.70) | 1146 (36.04) | 2035 (63.96) | |
| Yes | 5985 (65.30) | 3766 (62.92) | 2219 (37.08) | |
| **Community-level characteristics** | | | | |
| **Place of residence** | | | | <0.001 |
| Urban | 2010 (21.93) | 1362 (67.75) | 648 (32.25) | |
| Rural | 7156 (78.07) | 3550 (49.61) | 3606 (50.39) | |
| **Administrative divisions** | | | | <0.001 |
| Barishal | 507 (5.53) | 189 (37.3) | 318 (62.7) | |
| Chattogram | 1983 (21.63) | 1026 (51.75) | 957 (48.25) | |
| Dhaka | 2214 (24.16) | 1376 (62.16) | 838 (37.84) | |
| Khulna | 926 (10.11) | 662 (71.45) | 265 (28.55) | |
| Mymenshingh | 706 (7.70) | 241 (34.17) | 465 (65.83) | |
| Rajshahi | 1071 (11.69) | 613 (57.26) | 458 (42.74) | |
| Rangpur | 993 (10.84) | 494 (49.73) | 499 (50.27) | |
| Sylhet | 766 (8.35) | 310 (40.53) | 455 (59.47) | |

(*Continued*)

**Table 1.** (Continued)

| Variables | Total n (%) | Home delivery | | *P value* |
| --- | --- | --- | --- | --- |
| | | No; n (%) | Yes; n (%) | |
| **Community wealth status** | | | | <0.001 |
| Low | 4154 (45.32) | 1558 (37.52) | 2595 (62.48) | |
| High | 5012 (54.68) | 3354 (66.91) | 1659 (33.09) | |
| **Community women education level** | | | | <0.001 |
| Low | 4676 (51.01) | 1914 (40.93) | 2762 (59.07) | |
| High | 4490 (48.99) | 2999 (66.78) | 1492 (33.22) | |
| **Community exposure to media** | | | | |
| Low | 4498 (49.07) | 1822 (40.52) | 2675 (59.48) | <0.001 |
| High | 4668 (50.93) | 3090 (66.18) | 1579 (33.82) | |

CI = Confidence Interval

4–7 visits (AOR: 0.17, CI: 0.14–0.20), and 8 or above (AOR: 0.09, CI: 0.06–0.13) were less likely to give home delivery compared to those having no ANC visit. Compared to women whose last pregnancy was planned, those whose previous children were undesired had 1.18 times greater chances of giving birth at home (AOR: 1.18, CI: 1.04–1.33). There was a 16% and 30% decrease in the likelihood of home births among women from communities with high wealth status (AOR: 0.84, CI: 0.72–0.98) and high media exposure (AOR: 0.70, CI: 0.60–0.80), respectively. In the same way, those surveyed from communities where women had more educational attainment were 18% less likely to give birth at home (AOR: 0.82, CI: 0.71–0.94). Women from Khulna (AOR: 0.50, CI: 0.41–0.62) and Rajshahi divisions (AOR: 0.80, CI: 0.64–0.99) had lower probabilities of giving birth at home compared to those from the Dhaka division, indicating a considerable variation in home delivery between divisions (Table 2).

## Discussion

The main objective of this study was to map the prevalence of home delivery practice and to determine its associated correlates among women at their last birth in Bangladesh using the mixed-effect binary logistic regression model. The correlates of home delivery that were found to be significant were women's age and education level, household wealth status, ANC visit, last pregnancy intention, community women education level, community-level exposure to media, and community-level ANC visit.

When examining the district-level prevalence, home delivery was more common in Bangladesh's periphery districts. The regression model also showed a significant divisional variation in home delivery in Bangladesh. The populations studied were from a wide range of geographic areas with varying characteristics and social norms. Socioeconomic factors, health care coverage, accessibility, and the availability of high-quality maternal health services all have a significant impact on the choice of delivery location [37–39]. In this study, the highest prevalence was observed in the Bandarban district (84.58%), which is a hill tract region in Bangladesh. Shahabuddin et al. found similar results with young women in Nepal's mountainous areas vs those in the Terai area regarding the likelihood of institutional delivery [40]. This implies that health facility delivery will be challenging for the majority of Bangladeshi women who reside in the nation's impoverished areas unless there is an equitable distribution of health facilities and the removal of accessibility barriers, such as the provision of efficient and effective referral services.

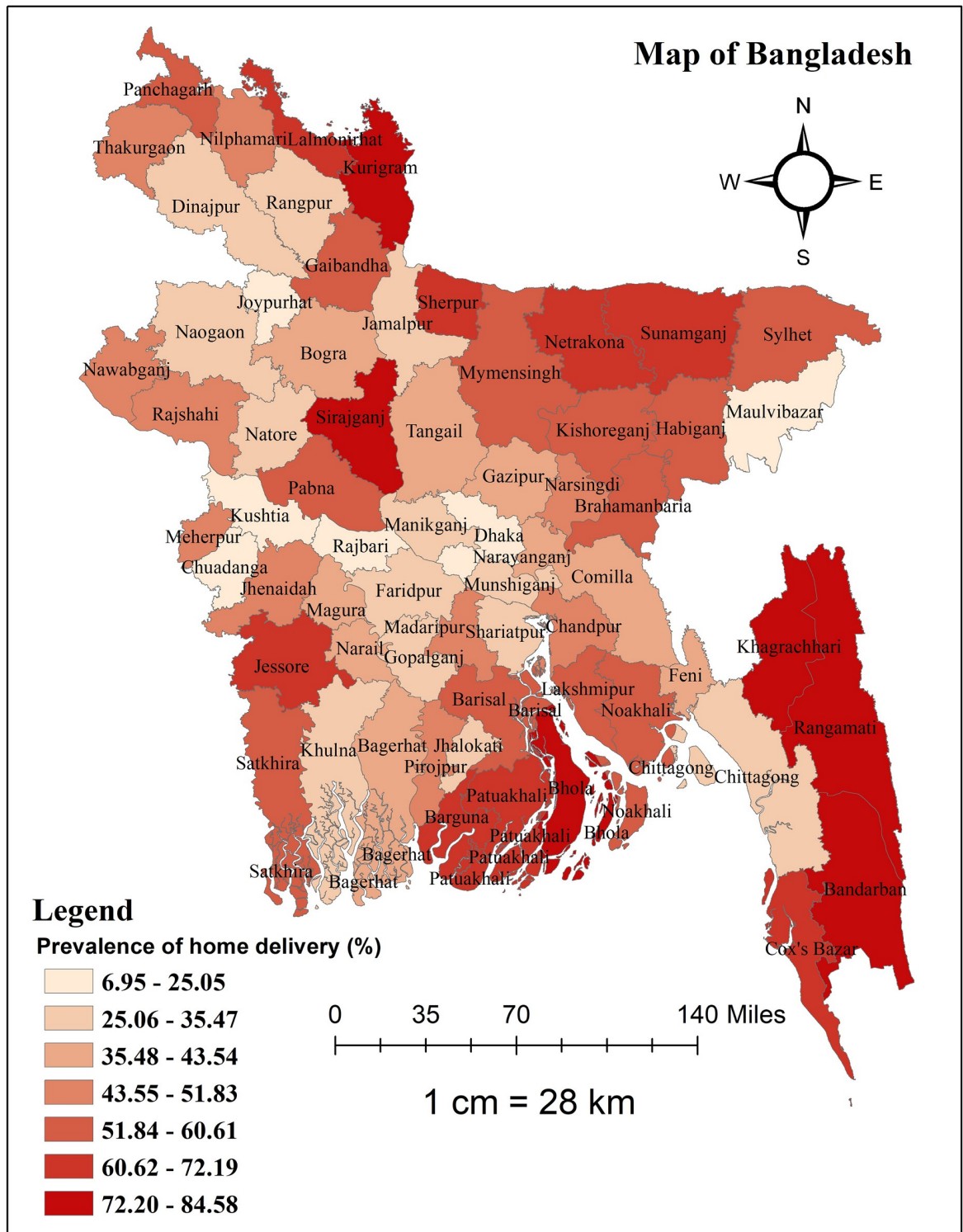

**Fig 2. Map showing the district-level distribution of prevalence of home delivery in Bangladesh.** This map was generated using data obtained from the MICS 2019 survey, with the base shapefile of Bangladesh from a freely available online source: https://data.humdata. org/dataset/cod-ab-bgd?.

**Table 2. Multilevel binary logistic regression analysis of factors associated with home delivery.**

| Variables | Model 0 | Model 1 | | Model 2 | | Model 3 | |
|---|---|---|---|---|---|---|---|
| | | AOR (95% CI) | *P* value | AOR (95% CI) | *P* value | AOR (95% CI) | *P* value |
| **Measures of association (fixed-effects)** | | | | | | | |
| **Individual-level factors** | | | | | | | |
| **Age of women** | | | | | | | |
| 15–19 years (Ref) | | 1 | | | | 1 | |
| 20–34 years | | 1.11(0.95–1.30) | 0.176 | | | 1.13(0.97–1.32) | 0.121 |
| 35–49 years | | 1.27(1.00–1.60) | 0.050 | | | 1.29(1.02–1.63) | 0.034 |
| **Women's education level** | | | | | | | |
| Pre-primary or none (Ref) | | 1 | | | | 1 | |
| Primary | | 0.82(0.66–1.02) | 0.076 | | | 0.89(0.71–1.10) | 0.283 |
| Secondary | | 0.48(0.39–0.59) | <0.001 | | | 0.58(0.46–0.72) | <0.001 |
| Higher secondary+ | | 0.25(0.20–0.33) | <0.001 | | | 0.31(0.24–0.41) | <0.001 |
| **Wealth status** | | | | | | | |
| Poorest (Ref) | | 1 | | | | 1 | |
| Second | | 0.67(0.57–0.79) | <0.001 | | | 0.73(0.62–0.86) | <0.001 |
| Middle | | 0.53(0.45–0.62) | <0.001 | | | 0.60(0.50–0.72) | <0.001 |
| Fourth | | 0.39(0.32–0.46) | <0.001 | | | 0.48(0.39–0.59) | <0.001 |
| Richest | | 0.26(0.21–0.32) | <0.001 | | | 0.31(0.24–0.40) | <0.001 |
| **Age at first marriage/union** | | | | | | | |
| ≤ 18 years (Ref) | | 1 | | | | 1 | |
| >18 years | | 0.95(0.84–1.08) | 0.466 | | | 0.88(0.77–0.99) | 0.039 |
| **ANC visit** | | | | | | | |
| No visit (Ref) | | 1 | | | | 1 | |
| 1–3 visit | | 0.36(0.31–0.43) | <0.001 | | | 0.38(0.33–0.45) | <0.001 |
| 4–7 visit | | 0.15(0.13–0.18) | <0.001 | | | 0.17(0.14–0.20) | <0.001 |
| 8 and above | | 0.08(0.06–0.12) | <0.001 | | | 0.09(0.06–0.13) | <0.001 |
| **Last pregnancy intention** | | | | | | | |
| Intended (Ref) | | 1 | | | | 1 | |
| Unintended | | 1.11(0.99–1.26) | 0.081 | | | 1.18(1.04–1.33) | 0.009 |
| **Exposure to media** | | | | | | | |
| No (Ref) | | 1 | | | | 1 | |
| Yes | | 0.68(0.60–0.76) | <0.001 | | | 0.89(0.77–1.02) | 0.085 |
| Community-level factors | | | | | | | |
| **Place of residence** | | | | | | | |
| Urban (Ref) | | | | 1 | | 1 | |
| Rural | | | | 1.35(1.16–1.56) | <0.001 | 0.99(0.85–1.17) | 0.926 |
| **Administrative divisions** | | | | | | | |
| Barishal | | | | 1.44(1.15–1.80) | 0.001 | 1.46(1.15–1.84) | 0.002 |
| Chattogram | | | | 1.50(1.27–1.79) | <0.001 | 1.47(1.23–1.76) | <0.001 |
| Dhaka (Ref) | | | | 1 | | 1 | |
| Khulna | | | | 0.48(0.39–0.58) | <0.001 | 0.50(0.41–0.62) | <0.001 |
| Mymenshingh | | | | 2.02(1.56–2.62) | <0.001 | 1.78(1.36–2.34) | <0.001 |
| Rajshahi | | | | 0.89(0.72–1.09) | 0.245 | 0.80(0.64–0.99) | 0.038 |
| Rangpur | | | | 1.10(0.90–1.34) | 0.355 | 1.21(0.98–1.49) | 0.070 |
| Sylhet | | | | 1.44(1.15–1.80) | 0.002 | 1.36(1.07–1.73) | 0.011 |
| **Community wealth status** | | | | | | | |
| Low (Ref) | | | | 1 | | 1 | |

*(Continued)*

**Table 2.** (Continued)

| Variables | Model 0 | Model 1 | | Model 2 | | Model 3 | |
|---|---|---|---|---|---|---|---|
| | | AOR (95% CI) | *P* value | AOR (95% CI) | *P* value | AOR (95% CI) | *P* value |
| High | | | | 0.46(0.41–0.53) | <0.001 | 0.84(0.72–0.98) | 0.026 |
| **Community women education** | | | | | | | |
| Low (Ref) | | | | 1 | | 1 | |
| High | | | | 0.48(0.43–0.54) | <0.001 | 0.82(0.71–0.94) | 0.004 |
| **Community exposure to media** | | | | | | | |
| Low (Ref) | | | | 1 | | 1 | |
| High | | | | 0.55(0.49–0.62) | <0.001 | 0.70(0.60–0.80) | <0.001 |
| **Measures of variation (random-effects)** | | | | | | | |
| Variance (SE) | 1.47 (0.117) | 0.52 (0.077) | | 0.46 (0.066) | | 0.41 (0.070) | |
| PCV | Ref | 64.63% | | 68.71% | | 72.11% | |
| ICC | 30.80% | 13.61% | | 12.23% | | 11.00% | |
| MOR | 3.16 | 1.98 | | 1.90 | | 1.84 | |
| **Model fitness** | | | | | | | |
| Log Likelihood | -6150.62 | -5106.42 | | -5599.34 | | -4984.28 | |
| AIC | 12305.25 | 10246.84 | | 11224.67 | | 10024.56 | |

→ Ref = Reference category, AOR = Adjusted Odds Ratio, CI = Confidence Interval.

→ Model 0 was the null model (only the intercept model) included no independent variable.

→ Model 1 includes only individual-level factors (mean variance inflation factor [VIF] = 1.98).

→ Model 2 includes only community-level factors (mean VIF = 1.42).

→ Model 3 includes both individual and community-level factors (mean VIF = 1.95).

SE = Standard Error, PCV = Proportional Change in Variance, ICC = Intra-Class Correlation, MOR = Median Odds Ratio, AIC = Akaike Information Criterion.

According to the age of mothers, women from the 35–49 age group had a higher probability of delivery at home compared to women who were aged between 15 and 19 years. Previous research from Tanzania [41] and Nepal [42, 43] also revealed consistent findings. These results collectively showed that older women were more likely to give birth at home than younger women. This outcome could be the result of older women believing they have enough expertise to deliver babies on their own without the help of trained professionals. But because they have no prior experience giving birth, young women often anticipate difficulties associated with pregnancy and childbirth [44].

A woman's likelihood of giving birth at home decreased with education. A greater level of education among women in the same community impacts their decision to give birth in a health facility, in addition to the favorable effects of individual education levels on their usage of health facilities for delivery. Similar results were also found in research carried out in Ghana [45], and Malawi [46], where the authors found that women who had finished secondary or higher education were less likely than those who had no formal education to give birth at home. According to a recent study, having education makes it more likely that a woman will choose to give birth in a hospital or maternity home rather than at home or somewhere else [47]. This may be due to because of education raising people's knowledge of health as a whole and exposing them to the advantages of complication prevention [47]. When considered collectively, these factors may motivate women to look for improved medical treatment, which may include giving birth in a hospital.

Compared to women from lower-income houses, we discovered that women from wealthier households were less likely to give birth at home. Our findings also align with earlier

research conducted in other LMICs such as Nepal [40, 48], Malawi [46], Ghana [45] and Guinea-Bissau [49]. Financial situations may have contributed to the difference in place of delivery between the affluent and the poor. When a poor woman needs to give birth at a healthcare facility, she may face financial difficulties due to the expense of transportation and other delivery-related expenses [45]. Additionally, women from higher socioeconomic groups with higher education and wealth status may be more empowered to make decisions for themselves, obtain information, and be financially independent enough to support themselves, travel to a medical facility and pay for services when needed, as well as to easily absorb health-related messages from the media and medical professionals [50, 51].

It is well known that the use of ANC affects mothers' decisions about where to give birth, with ANC users often favoring institutional deliveries under the supervision of health professionals [52, 53]. Thus, it was not surprising that woman who had no ANC visits had a greater rate of home birth than those who had at least one ANC visit in the current research. It is shown that receiving enough ANC can increase a pregnant woman's awareness of probable challenges and safe delivery techniques, which will motivate her to give birth in an institution [53, 54]. Furthermore, it has been argued that women who visit medical facilities for ANC check-ups could get guidance and counseling from medical staff [55]. Both instances educate them regarding the risks associated with home delivery. It is also argued that women who have received important information during ANC may choose to give birth in a healthcare facility as a safeguard against unanticipated difficulties that may arise with a home delivery [55]. Social networks are expected to be the medium via which women who have received ANC within a particular community share their knowledge. This information sharing might then encourage women living nearby to look for better healthcare options, such as choosing facility-based births. This phenomenon raises the possibility that community-wide maternal health practices may be impacted by ANC use [50].

Although unintended pregnancies have been linked to pregnancy-related complications like poor weight gain, pregnancy-induced hypertension, and anemia that require hospital delivery [56, 57], women in this study who had unintended pregnancies were more likely to give birth at home than those whose pregnancies were planned. This supports the findings of earlier research [58, 59]. Given this situation, the high prevalence of home births attributable to unplanned pregnancies may be explained by the sociocultural stigma and restrictions that prevent some women from accessing maternal healthcare services, including facility deliveries [60, 61]. Furthermore, the results highlight the significance of encouraging pregnant women about the risks of home delivery for unplanned pregnancies in order to encourage them to have facility delivery [19].

We found that higher levels of community exposure to media significantly reduced the odds of home delivery. Given that women have access to more health information and may obtain knowledge from the media, this is not unexpected as they are more likely to make informed decisions. These might help mothers by providing them with the information they need to seek out better maternal healthcare services [62]. The phenomenon may be explained by the fact that the majority of media outlets frequently promote institutional delivery, which may persuade mothers to adopt favorable attitudes toward giving birth in a health facility [50].

## Strengths and limitations

The application of multilevel regression analysis, which enabled the investigation of both individual and community-level factors impacting home delivery, was one of the study's strengths. Large sample sizes were also used in the study, which improved the findings' generalizability to Bangladesh's larger population. However, there are some limitations to consider. First of all,

because the study depended on self-reported data, it might be biased toward social desirability and recollection. Second, because the data are cross-sectional, it is more difficult to demonstrate causation and ascertain the time course of the association between the variables and home delivery. Longitudinal studies would provide more robust evidence in this regard. Finally, the study did not explore certain potential factors, such as cultural beliefs and attitudes towards home delivery, which could have influenced the findings.

## Conclusion

In Bangladesh, home births accounted for over half of all births, where women with greater levels of education, affluence, and ANC visits had a much lower rate of home deliveries, but women in the 35–49 age range, and who had an unplanned pregnancy experienced a higher rate. Target-specific interventions aimed at reducing home births should prioritize addressing disparities related to maternal education, family socioeconomic status, media access, and closing the wealth gap between affluent and poor households as well as between rural and urban locations. The results of this study might help Bangladeshi stakeholders who are in charge of maternal and child healthcare in order to plan interventions that would decrease home births and improve maternity care facilities during delivery. The Government need to think about making investments in creative strategies to increase pregnant women's access to healthcare facilities. To decrease home delivery in Bangladesh, more subsidies or easier access to free services for institutional delivery could be useful tactics.

The study's conclusions may have a significant impact on interventions and policy decisions that are proposed for specific agencies and ministries in Bangladesh to lower the prevalence of home birth in Bangladesh. The Directorate General of Health Services (DGHS) under the Ministry of Health and Family Welfare along with different non-government organizations should spearhead awareness campaigns on the benefits of skilled birth attendance and the risks of home delivery. Concurrently, the Ministry of Women and Children Affairs, in collaboration with the Ministry of Education, should focus on improving education and awareness among women regarding the benefits of skilled birth attendance and the potential risks associated with home delivery. Additionally, interventions should address socioeconomic barriers by providing financial support for transportation and improving the affordability of maternal health services. The DGHS, Bangladesh should also enhance the availability and quality of healthcare facilities in rural areas in order to curve the reliance on home delivery in these regions.

## Supporting information

**S1 File.**
(ZIP)

## Acknowledgments

We would like to show our gratitude to the Multiple Indicator Cluster Survey (MICS-2019) Program for providing data access used in this research. We would also like to gratefully acknowledge the study's participants, reviewers and the academic editors of our manuscript.

## Author Contributions

**Conceptualization:** Sarmistha Paul Setu, U. K. Majumder.

**Data curation:** Rakhi Dey, Satyajit Kundu.

**Formal analysis:** Rakhi Dey, Susmita Rani Dey, Meem Haque, Anushuya Binta Rahman, Satyajit Kundu, Sarmistha Paul Setu.

**Investigation:** Rakhi Dey, Satyajit Kundu, Sarmistha Paul Setu.

**Methodology:** Rakhi Dey, Susmita Rani Dey, Meem Haque, Anushuya Binta Rahman, Satyajit Kundu, Sarmistha Paul Setu, U. K. Majumder.

**Resources:** Rakhi Dey.

**Software:** Rakhi Dey, Susmita Rani Dey, Meem Haque, Anushuya Binta Rahman, Satyajit Kundu.

**Supervision:** Sarmistha Paul Setu, U. K. Majumder.

**Validation:** Satyajit Kundu, U. K. Majumder.

**Visualization:** Satyajit Kundu, Sarmistha Paul Setu, U. K. Majumder.

**Writing – original draft:** Rakhi Dey, Susmita Rani Dey, Meem Haque, Anushuya Binta Rahman, Satyajit Kundu, Sarmistha Paul Setu, U. K. Majumder.

**Writing – review & editing:** Rakhi Dey, Susmita Rani Dey, Meem Haque, Anushuya Binta Rahman, Satyajit Kundu, Sarmistha Paul Setu, U. K. Majumder.

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
