## [Decision Letter · Decision Letter 0]

3 Apr 2024

PONE-D-23-36533Mapping the prevalence and Covariates Associated with Home Delivery in Bangladesh: A Multilevel Regression AnalysisPLOS ONE

Dear Dr. Majumder,

Thank you for submitting your manuscript to PLOS ONE. After careful consideration, we feel that it has merit but does not fully meet PLOS ONE’s publication criteria as it currently stands. Therefore, we invite you to submit a revised version of the manuscript that addresses the points raised during the review process.

We look forward to receiving your revised manuscript.

Kind regards,

Mohammad Nayeem Hasan

Academic Editor

PLOS ONE

Journal Requirements:

3. We note that Figure 2 in your submission contain map image which may be copyrighted. All PLOS content is published under the Creative Commons Attribution License (CC BY 4.0), which means that the manuscript, images, and Supporting Information files will be freely available online, and any third party is permitted to access, download, copy, distribute, and use these materials in any way, even commercially, with proper attribution. For these reasons, we cannot publish previously copyrighted maps or satellite images created using proprietary data, such as Google software (Google Maps, Street View, and Earth). For more information, see our copyright guidelines: http://journals.plos.org/plosone/s/licenses-and-copyright.

Reviewers' comments:

Reviewer's Responses to Questions

**Comments to the Author**

1. Is the manuscript technically sound, and do the data support the conclusions?

Reviewer #1: Yes

Reviewer #2: Yes

2. Has the statistical analysis been performed appropriately and rigorously? 

Reviewer #1: Yes

Reviewer #2: Yes

3. Have the authors made all data underlying the findings in their manuscript fully available?

Reviewer #1: Yes

Reviewer #2: Yes

4. Is the manuscript presented in an intelligible fashion and written in standard English?

Reviewer #1: Yes

Reviewer #2: Yes

5. Review Comments to the Author

Reviewer #1: Dear Authors,

I hope this email finds you well. I wanted to let you know that I found your article to be well-written, covering an interesting and helpful topic that could prove valuable for future recommendations and actions.

I strongly recommend that the article be published with no significant changes. However, I would like to suggest updating the reference list, as approximately 40% of the sources listed are dated before 2015.

Thank you for taking the time to consider my feedback.

Best regards,

Reviewer #2: (Line 100 : skilled “attendance” is it a typo error of attendant.

Line 132 : ever-married : what is the meaning of it ?

Line 131 : and study procedure can be found elsewhere , it should be replaced with proper place as author is given reference for it .

Link 148 : wealth status (poorest, poorer, middle, richer, richest)… Representation doesn’t look here , it can be replaced with income range , like low income (0 – 100 USD per week/month ,100 – 500 USD ) so on.

149 : exposure to media .What is the role of exposure of media in this study?

Classification should be modify or add reference where from it was adopted words like

“Poorest” and “Richest”, in my concern it should be replaced with the range of income.

Line 189 : Result section Define this line “9,166 (weighted) women who had at least one live delivery in the two years before to 190 the survey were part of this investigation.” while shading some light of the data given in the table 1 section Age at first marriage/union , ≤ 18 years , number of correspondents is 6632 . As 7069 correspondents were of 20 – 34 year range. As well as 6632 were less than 18 year.

Please explain it , I am confused now .

Line 292 and line 295 is seems the repetition of the same statement in a different way . Rewrite it please.

Line 286 and line 300 also looks similar, better this sentence should be used in the later part of discussions to make it general statement with combining the effect of education and wealth status.

Check the spellings and reference throughout use correct format , As ref 14, 16 and 22 . Rest if possible to support study some recent references also can be added. )

6. PLOS authors have the option to publish the peer review history of their article (what does this mean?). If published, this will include your full peer review and any attached files.

Reviewer #1: **Yes: **Beisan A. Mohammad,PhD, MSc Pharm Sci, BPharm, MSc MEd

Reviewer #2: No

---

## [Author Response · Author response to Decision Letter 0]

27 Apr 2024

Response to the Reviewers' comments:

Reviewer #1: 

Comment: Dear Authors,

I hope this email finds you well. I wanted to let you know that I found your article to be well-written, covering an interesting and helpful topic that could prove valuable for future recommendations and actions. I strongly recommend that the article be published with no significant changes. However, I would like to suggest updating the reference list, as approximately 40% of the sources listed are dated before 2015. Thank you for taking the time to consider my feedback.

Author’s Response: Dear reviewer, thank you for your appreciation and time to review our paper. We agree with your argument that almost 40% of references are dated before 2015. For your kind information, we found a few references that are available on the home delivery, rather most of the recent works focused mainly on the institutional delivery. However, we have updated some references this time according to your suggestion and also incorporated some recent references that worked on home delivery. Please check the reference section.

Besides that, we had to keep some references dated even before 2010 due to supporting some arguments in our problem statement and discuss some results of our study. We found those references are crucial to rationalize our statements. So, we hope you will also agree with us; however, if you suggest some references that are strongly required to improve our paper, we will be happy to include them.

Reviewer #2:

Comment: Line 100: skilled “attendance” is it a typo error of attendant.

Author’s Response: Dear reviewer, thank you for your time to review our paper. We corrected the typo you pointed out.

Comment: Line 132: ever-married: what is the meaning of it?

Author’s Response: We have defined the term “ever-married” with reference as follows:

“The ever-married women were those who had been married at least once in their lives though they may not be currently married [26]”

Comment: Line 131: and study procedure can be found elsewhere, it should be replaced with proper place as author is given reference for it.

Author’s Response: We have replaced it with the report of MICS 2019. Now it reads:

“The detailed information on sampling technique, questionnaire, and study procedure can be found in the MICS 2019 report [25]”

Comment: Line 148: wealth status (poorest, poorer, middle, richer, richest). Representation doesn’t look here, it can be replaced with income range, like low income (0–100 USD per week/month, 100–500 USD ) so on.

Author’s Response: Thank you for your comment. MICS measures the household wealth status based on ownership of different household assets. In Bangladesh MICS 2019, 25 (groups of) variables that were used for the construction of the Bangladesh Wealth Index. The wealth index is assumed to capture the underlying long-term wealth through information on the household assets and is intended to produce a ranking of households by wealth, from poorest to richest. The wealth index does not provide information on absolute poverty, current income or expenditure levels. Each household in the total sample is then assigned a wealth score based on the assets owned by that household and on the final factor scores obtained as described above. The survey household population is then ranked according to the wealth score of the household they are living in and is finally divided into 5 equal parts (quintiles) from lowest (poorest) to highest (richest). In MICS report, there is a separate variable for wealth status that we used where the quintiles of wealth status were named as poorest, second, middle, fourth, and richest. We also replaced “poorer” with “second”, and “richer” with “fourth” in this study (please see in the method section). The aforementioned description are taken from the MICS 2019 report. There is no estimation of the income amount (like in USD) available in the MICS data sets. Previous studies also used this variable as wealth quintile. Please see a previous study and MICS report 2019 below:

Bangladesh Bureau of Statistics (BBS) and UNICEF Bangladesh. Progotir Pathey, Bangladesh Multiple Indicator Cluster Survey 2019, Survey Findings Report. Dhaka, Bangladesh: Bangladesh Bureau of Statistics (BBS); 2019.

Chowdhury TR, Chakrabarty S, Rakib M, Winn S, Bennie J. Risk factors for child stunting in Bangladesh: an analysis using MICS 2019 data. Arch Public Heal. 2022;80:1–12.

We also added a statement with reference in the method section in this study as follows:

“The household wealth status was calculated based on the ownership of different household assets using principal component analysis (PCA) [28].”

Comment: Line 149: exposure to media . What is the role of exposure of media in this study?

Author’s Response: The Bangladesh MICS, 2019 collected information on exposure to mass media, where the information was collected on exposure to newspapers / magazines, radio and television among women 15-49 years. We categorized women having exposure to media if they had access to either reading newspapers / magazines, or listening radio or watching television following the guidelines of MICS and previous literature (please see ref below). 

Saleheen AAS, Afrin S, Kabir S, Habib MJ, Zinnia MA, Hossain MI, et al. Sociodemographic factors and early marriage among women in Bangladesh, Ghana and Iraq: An illustration from Multiple Indicator Cluster Survey. Heliyon. 2021;7. 

We added the following statement in the method section now:

“We classified women as having exposure to media if they had access to read newspapers or magazines, listen to the radio, or watch television.”

In response to your question- “what is the role of exposure of media in this study”, we included this variable based on the previous studies where media exposure was found to be associated with the place of delivery. Though we didn’t find any significant association for the individual level exposure to media, a significant association of community level media exposure was significantly associated with the home delivery in this study, and we discussed this finding.

Comment: Classification should be modified or add reference where from it was adopted words like “Poorest” and “Richest”, in my concern it should be replaced with the range of income.

Author’s Response: We have provided the references of MICS 2019 report, where the wealth status also categorized as quintile (poorest, second, middle, fourth, richest). Please see ref 25.

Comment: Line 189: Define this line “9,166 (weighted) women who had at least one live delivery in the two years before to 190 the survey was part of this investigation.” 

Author’s Response: We have rewritten the line as follows:

“A total of 9,166 (weighted) women who had at least one live delivery in the two years preceding the survey were included in this study.”

Comment: While shading some light of the data given in the Table 1 section Age at first marriage/union, ≤18 years, and number of correspondents is 6632. As 7069 correspondents were of 20–34 -year range. As well as 6632 were less than 18 year. Please explain it, I am confused now.

Author’s Response: These are two distinct variables. The variable “Age of women” represents the current age of women at the time of survey, where we found 7069 women were from 20-34 years age group. On the contrary, the variable “Age at first marriage/union” provides the information on about how old the woman was at the time of first marriage / union? Hence, the variable “Age at first marriage/union” provides the prior information, not current, and consequently, it shows that 7069 respondents were of 20–34 -year range, but 6632 were less than 18 year at their first marriage / union.

Comment: Line 292 and line 295 is seems the repetition of the same statement in a different way. Rewrite it please.

Author’s Response: Thank you for pointing this out. We have removed the 2nd line.

Comment: Line 286 and line 300 also looks similar, better this sentence should be used in the later part of discussions to make it general statement with combining the effect of education and wealth status.

Author’s Response: Thank you for this comment. We have placed this sentence at the end of the paragraph discussing association between socioeconomic status and home delivery in a single statement with combining the effect of education and wealth status. Now it reads:

“Additionally, women from higher socioeconomic group with higher education and wealth status may be more empowered to make decisions for themselves, obtain information, and be financially independent enough to support themselves, travel to a medical facility and pay for services when needed, as well as to easily absorb health-related messages from the media and from medical professionals [46,47]”

Comment: Check the spellings and reference throughout use correct format, as ref 14, 16 and 22. Rest if possible to support study some recent references also can be added.

Author’s Response: We followed the PLOS style for referencing. We have added some recent references also. Please see refs 18 to 24.

---

## [Decision Letter · Decision Letter 1]

25 Jun 2024

PONE-D-23-36533R1Mapping the prevalence and Covariates Associated with Home Delivery in Bangladesh: A Multilevel Regression AnalysisPLOS ONE

Dear Dr. Majumder,

Thank you for submitting your manuscript to PLOS ONE. After careful consideration, we feel that it has merit but does not fully meet PLOS ONE’s publication criteria as it currently stands. Therefore, we invite you to submit a revised version of the manuscript that addresses the points raised during the review process.

We look forward to receiving your revised manuscript.

Kind regards,

Mohammad Nayeem Hasan

Academic Editor

PLOS ONE

Reviewers' comments:

Reviewer's Responses to Questions

**Comments to the Author**

1. If the authors have adequately addressed your comments raised in a previous round of review and you feel that this manuscript is now acceptable for publication, you may indicate that here to bypass the “Comments to the Author” section, enter your conflict of interest statement in the “Confidential to Editor” section, and submit your "Accept" recommendation.

Reviewer #3: (No Response)

2. Is the manuscript technically sound, and do the data support the conclusions?

Reviewer #3: Partly

3. Has the statistical analysis been performed appropriately and rigorously? 

Reviewer #3: Yes

4. Have the authors made all data underlying the findings in their manuscript fully available?

Reviewer #3: Yes

5. Is the manuscript presented in an intelligible fashion and written in standard English?

Reviewer #3: Yes

6. Review Comments to the Author

Reviewer #3: Comments

Generally, this paper is essential to maternal and child health. The authors dealt with the phenomenon with a clear methodological step. I congratulate them on that. It’s great work, congratulations to the authors. However, they should pay attention to these comments and work on them to improve the paper.

Lines 1 and 2, the title reads: “Mapping the prevalence and Covariates Associated with Home Delivery in Bangladesh: A Multilevel Regression Analysis”. For consistency’s sake, they could consider capitalizing “prevalence” as the major keywords started with capital letters.

Line 41 to 43, they started the sentence with “though”. Technically, Abstract should be written in abstracting terms. As such, authors could consider removing “though”, and “however” from the abstract.

In line 43, the authors stated that “However, reasons behind this need to be explored in community-level”. the use of “reasons” appears as if the authors are going to present real reasons. Meanwhile, the work is quantitative. Quantitative language should be used here instead of qualitative terms/words. Preferably, they can use terms such as “predictors”, “plausible factors”, “determinants” and others. “Reasons” sounds more in a qualitative sense.

Line 51, the authors stated that “The overall weighted prevalence of home delivery was 46.41%”. it is appropriate to add the confidence intervals for this overall prevalence.

Line 51-58, the authors only presented fixed effects results, it will be appropriate if they capture the random effects results as well here, briefly.

In line 68, the authors included “delivery care” as a keyword and I wonder how that can be a keyword in this study.

In lines 76-77, the authors stated that “In most, but not all, nations during the past few decades, there has been a marked decline in home births”. This is a good point though; however, I was expecting a recent citation to align with this statement. However, the authors cited 1985, 1997. This is several decades ago as we are in 2024.

Lines 78-80, Reference 11 and 12 are citations in 2006 and 2012 on Maternal mortality trends. This is less acceptable as recent estimates of maternal mortality are available for use. Authors should consider revising references that are too old in the entire background section.

In line 108, the authors stated “Bangladesh has implemented several initiatives to address maternal health……..”. However, they didn’t elaborate on these initiatives. The section will read better and scientifically good if the authors narrate/synthesize the strategies/initiatives, adopted in Bangladesh, the success made by such strategies, and the failures if any. They have to do such a comparative analysis.

In lines 108 to 111, the authors presented “Issues such as inadequate healthcare infrastructure, geographical barriers, limited access to skilled birth attendants, and socioeconomic disparities continue to affect maternal health outcomes”. These are probable factors that might have been identified by earlier studies so to avoid plagiarism, authors have to reference this statement accordingly.

Lines 111 to 113, reads “Our study’s findings help in identifying the obstacles to health facility delivery and the variables influencing maternal fatalities during in-home birth in Bangladesh”. Is this a justification or statement of the problem? Authors should come clearly and state the problem and the justification for the study separately. They should not combine the problem statement and justification.

Authors should present the study design first before talking about the data source in the methods section.

In lines 136 to 142, the authors didn’t talk about “others” in the responses. Were there “others” in the responses given by the respondents and yes, how are “others” managed in this study?

Line 148, ANC visits were classified as no visits, 1-3 visits, and 4 or above visits. What informs this classification? Currently, the practice is less than 8 visits as poor and 8 or more as desirable. It could be appropriate if they reclassify ANC according to current standards. A similar thing applies to the community ANC variable. Here, authors classified them into less than 50% having 4 or more ANC visits which is even not consistent with their earlier ANC classification at the individual level. they should reconcile this.

Line 149, the authors should consider explaining how they generated exposure to mass media as the dataset used does not have a variable called mass media, to the best of my knowledge.

Line 249, the authors should check this grammar “was to mapping” and correct it appropriately, “The main objective of this study was to mapping the prevalence of home delivery practice”.

In the policy implication section, authors have to be specific. Who or which agency or ministry should implement what? They should direct their policy implications to specific agencies or ministries/departments of Bangladesh. For instance, targeted efforts by the Ministry of Health, Reproductive Unit or Public Health Division of Bangladesh should focus on……

Authors should consider discussing policy implications after the conclusion. So, they should shift policy implication to the last section after the conclusion.

Finally, authors should check for grammatical errors and if any exist, correct them. Thank you.

7. PLOS authors have the option to publish the peer review history of their article (what does this mean?). If published, this will include your full peer review and any attached files.

Reviewer #3: No

---

## [Author Response · Author response to Decision Letter 1]

3 Aug 2024

Response to the Reviewers' comments:

Comment: Generally, this paper is essential to maternal and child health. The authors dealt with the phenomenon with a clear methodological step. I congratulate them on that. It’s great work, congratulations to the authors. However, they should pay attention to these comments and work on them to improve the paper.

Author’s response: Dear Reviewer, Thank you for the time and for considering our paper for reviewing. We appreciate your effort and are thankful for your appreciation and valuable feedback.

Comment: Lines 1 and 2, the title reads: “Mapping the prevalence and Covariates Associated with Home Delivery in Bangladesh: A Multilevel Regression Analysis”. For consistency’s sake, they could consider capitalizing “prevalence” as the major keywords started with capital letters.

Author’s response: Thank you for pointing this out. We have capitalized it in the title.

Comment: Line 41 to 43, they started the sentence with “though”. Technically, Abstract should be written in abstracting terms. As such, authors could consider removing “though”, and “however” from the abstract.

Author’s response: We have revised the introduction section of the abstract and removed ‘though’ and ‘however’ from the abstract. Now it reads:

Bangladesh has made an intense effort to improve maternal healthcare facilities including facility delivery, but the number of home deliveries is still very high Therefore, this study aims to find out district-wise prevalence and determine the individual and community-level covariates related to home delivery among women in Bangladesh.

Comment: In line 43, the authors stated that “However, reasons behind this need to be explored in community-level”. the use of “reasons” appears as if the authors are going to present real reasons. Meanwhile, the work is quantitative. Quantitative language should be used here instead of qualitative terms/words. Preferably, they can use terms such as “predictors”, “plausible factors”, “determinants” and others. “Reasons” sounds more in a qualitative sense.

Author’s response: Thank you for this insightful comment. We have changed the sentence and revised the section. Please see the previous response.

Comment: Line 51, the authors stated that “The overall weighted prevalence of home delivery was 46.41%”. it is appropriate to add the confidence intervals for this overall prevalence.

Author’s response: Thank you for your suggestion. We have added the confidence intervals for the prevalence in the abstract.

Comment: Line 51-58, the authors only presented fixed effects results, it will be appropriate if they capture the random effects results as well here, briefly.

Author’s response: We have added a statement from the random effects results in the abstract. Please see the statement below:

The intercept-only regression model demonstrates that the likelihood of women from various clusters having home delivery varied significantly (variance: 1.47, standard error [SE]: 0.117), indicating the applicability of multilevel regression modeling.

Comment: In line 68, the authors included “delivery care” as a keyword and I wonder how that can be a keyword in this study.

Author’s response: We have removed ‘delivery care’ from the keywords.

Comment: In lines 76-77, the authors stated that “In most, but not all, nations during the past few decades, there has been a marked decline in home births”. This is a good point though; however, I was expecting a recent citation to align with this statement. However, the authors cited 1985, 1997. This is several decades ago as we are in 2024.

Author’s response: We have updated these with some recent references (Please check refs 6-8).

Comment: Lines 78-80, Reference 11 and 12 are citations in 2006 and 2012 on Maternal mortality trends. This is less acceptable as recent estimates of maternal mortality are available for use. Authors should consider revising references that are too old in the entire background section.

Author’s response: Dear reviewer, Thank you for your observation. We have tried to update the references for the entire background section. The references you are referring to, was to support the following statement “The expansion of institutional delivery coverage and the use of skilled birth attendants during deliveries are only a couple of the measures that have been put forth to lower this maternal, fetal, and newborn mortality.” To support the maternal mortality rate and trends, we used several recent references those are mostly after 2019. Please check the refs 11-14.

Comment: In line 108, the authors stated “Bangladesh has implemented several initiatives to address maternal health……..”. However, they didn’t elaborate on these initiatives. The section will read better and scientifically good if the authors narrate/synthesize the strategies/initiatives, adopted in Bangladesh, the success made by such strategies, and the failures if any. They have to do such a comparative analysis.

Author’s response: We have added some statements demonstrating the available health services and initiatives of Bangladesh with references. Now it reads:

Bangladesh has implemented several initiatives related to maternal health services to address maternal health. The key maternal health services available in Bangladesh include antennal care, post-abortion care, basic essential obstetric care (EOC), comprehensive EOC, postnatal care for mothers, intensive care unit for mothers, introduction of health vouchers scheme for poor women, deployment of community-based skilled birth attendants, and introduction of the midwifery programme [28–30]. But not all health facilities, particularly those in rural regions in Bangladesh, offer all kinds of maternity and newborn health services [29]. Consequently, despite progress, Bangladesh still faces challenges in reducing maternal mortality [31].

Comment: In lines 108 to 111, the authors presented “Issues such as inadequate healthcare infrastructure, geographical barriers, limited access to skilled birth attendants, and socioeconomic disparities continue to affect maternal health outcomes”. These are probable factors that might have been identified by earlier studies so to avoid plagiarism, authors have to reference this statement accordingly.

Author’s response: We already added the references for this statement. Please see refs 31 & 32.

Comment: Lines 111 to 113, reads “Our study’s findings help in identifying the obstacles to health facility delivery and the variables influencing maternal fatalities during in-home birth in Bangladesh”. Is this a justification or statement of the problem? Authors should come clearly and state the problem and the justification for the study separately. They should not combine the problem statement and justification.

Author’s response: We have revised the sentence now and focused on the objective of the study. We have separated the problem statement and justification of the study now. 

Comment: Authors should present the study design first before talking about the data source in the methods section.

Author’s response: We have re-organized the section, keeping study design before the data source in the method section as per reviewers comment.

Comment: In lines 136 to 142, the authors didn’t talk about “others” in the responses. Were there “others” in the responses given by the respondents and yes, how are “others” managed in this study?

Author’s response: We have clarified about the other’s home now. In the question, there were two options for home delivery along with the name of some other institutions. The two options for home delivery were “respondent’s home” and “other’s home”, where the other’s home meant the home of neighbours, friends, relatives or any birth attendants, and we have clarified this in the method section now as follows:

The other’s home referred to the home of neighbors, friends or relatives of the respondents, or any birth attendant’s home.

Comment: Line 148, ANC visits were classified as no visits, 1-3 visits, and 4 or above visits. What informs this classification? Currently, the practice is less than 8 visits as poor and 8 or more as desirable. It could be appropriate if they reclassify ANC according to current standards. A similar thing applies to the community ANC variable. Here, authors classified them into less than 50% having 4 or more ANC visits which is even not consistent with their earlier ANC classification at the individual level. they should reconcile this.

Author’s response: Thank you for your insightful comment. In response, we have recategorized the individual-level ANC visits into four categories: no visits, 1-3 visits, 4-7 visits, and 8 or more visits. This reclassification aligns with the current WHO recommendations for antenatal care (ANC) visits. Following this adjustment, we have reanalyzed the data accordingly.

Regarding the community-level ANC visit variable, we encountered a significant limitation. When we attempted to create a separate community-level variable based on the criterion of 8 or more visits as indicative of high community-level ANC, we observed that less than 1% of participants fell into this category. This extremely low prevalence rendered the variable ineffective for meaningful analysis and interpretation.

Consequently, we have excluded the community-level ANC visit variable from our study because of low percentage (less than 1%) in one category of two. We appreciate your feedback, which has been instrumental in refining our analysis. Please refer to the revised tables for the updated results.

Comment: Line 149, the authors should consider explaining how they generated exposure to mass media as the dataset used does not have a variable called mass media, to the best of my knowledge.

Author’s response: We have clarified how we generated the media exposure variable. Now it reads:

Media exposure was created as a dichotomous variable. Women were categorized as having media exposure if they reported access to at least one of the following information sources like reading newspapers/magazines, listening radio, or watching television. Those with access to any of these sources were classified as ‘yes’ for media exposure, while those without access to any were classified as ‘no’ [38].

Comment: Line 249, the authors should check the grammar “was to mapping” and correct it appropriately, “The main objective of this study was to mapping the prevalence of home delivery practice”.

Author’s response: Thank you for pointing this out. We have corrected this and fixed the grammatical errors throughout the manuscript.

Comment: In the policy implication section, authors have to be specific. Who or which agency or ministry should implement what? They should direct their policy implications to specific agencies or ministries/departments of Bangladesh. For instance, targeted efforts by the Ministry of Health, Reproductive Unit or Public Health Division of Bangladesh should focus on……

Author’s response: Thank you for this insightful comment. We have revised the recommendations and now it reads:

The study's conclusions may have a significant impact on interventions and policy decisions that are proposed for specific agencies and ministries in Bangladesh to lower the prevalence of home birth in Bangladesh. The Directorate General of Health Services (DGHS) under the Ministry of Health and Family Welfare along with different non-government organizations should spearhead awareness campaigns on the benefits of skilled birth attendance and the risks of home delivery. Concurrently, the Ministry of Women and Children Affairs, in collaboration with the Ministry of Education, should focus on improving education and awareness among women regarding the benefits of skilled birth attendance and the potential risks associated with home delivery. Additionally, interventions should address socioeconomic barriers by providing financial support for transportation and improving the affordability of maternal health services. The DGHS, Bangladesh should also enhance the availability and quality of healthcare facilities in rural areas in order to curve the reliance on home delivery in these regions.

Comment: Authors should consider discussing policy implications after the conclusion. So they should shift policy implication to the last section after the conclusion.

Author’s response: We have shifted the policy implication to the last section after the conclusion.

Comment: Finally, authors should check for grammatical errors and if any exist, correct them. Thank you.

Author’s response: We have thoroughly checked and fixed the grammatical errors throughout the manuscript this time.

---

## [Editor Report · Decision Letter 2]

29 Oct 2024

Mapping the prevalence and Covariates Associated with Home Delivery in Bangladesh: A Multilevel Regression Analysis

PONE-D-23-36533R2

Dear Dr. Majumder,

We’re pleased to inform you that your manuscript has been judged scientifically suitable for publication and will be formally accepted for publication once it meets all outstanding technical requirements.

Kind regards,

Md Hasinur Rahaman Khan, Ph.D.

Academic Editor

PLOS ONE

Additional Editor Comments (optional):

I have reviewed the response letter and happy with the responses. The authors need to refer the following multilevel modelling implemented to Bangladesh data in the text of the manuscript.

Multilevel Logistic Regression Analysis Applied to Binary Contraceptive Prevalence Data

Journal of Data Science, Vol. 9, pp. 93-110, 2011
---

## [Editor Report · Acceptance letter]

1 Nov 2024

PONE-D-23-36533R2 

PLOS ONE

Dear Dr. Majumder, 

I'm pleased to inform you that your manuscript has been deemed suitable for publication in PLOS ONE. Congratulations! Your manuscript is now being handed over to our production team.

Kind regards, 

on behalf of

Dr. Md Hasinur Rahaman Khan 

Academic Editor

PLOS ONE